# Comparison of Physicochemical, Microbiological Properties and Bioactive Compounds Content of Grassland Honey and other Floral Origin Honeys

**DOI:** 10.3390/molecules24162932

**Published:** 2019-08-13

**Authors:** Laura Agripina Scripcă, Liliana Norocel, Sonia Amariei

**Affiliations:** Faculty of Food Engineering, Stefan cel Mare University of Suceava, 720229 Suceava, Romania

**Keywords:** grassland honey, bioactive compounds, sugar composition, phenolic compounds

## Abstract

The aim of this study was to compare the physicochemical, the microbiological, and the antioxidant characteristics of unifloral honey, polyfloral honey, honeydew, and hay meadows honey. Hay meadow is type of semi-natural grassland with a great floral diversity, an important resource for pollinators. Grasslands are the source of the spring nectar honey obtained in May and June. Water content, sugars (fructose, glucose, sucrose, trehalose, melezitose, maltose, erlose, turanose, and raffinose), electrical conductivity, phenolic content (gallic acid, protocatechuic acid, 4-hydrxybenzoic acid, vanilic acid, chlorogenic acid, caffeic acid, p-coumaric acid, rosmarinic acid, myricetin, quercitin, luteolin, kaempferol), color, viscosity, and microbiological characteristics were performed for all samples of honey. The total polyphenols content was significant for grassland honey (21.50 mg/100 g) and honeydew (30.49 mg/100 g) and less significant for acacia (0.08 mg/100 g) and rape honey (0.14 mg/100 g). All samples were microbiologically safe, and standard plate count (SPC) values were <10 cfu/g for all the samples, but the grassland honey had the highest microbiological quality: 33.3% of samples without microorganisms, 50.0% with the presence of yeast under limit, and 16.7% with yeast and mold under limit, a situation that does not meet other types of honey. The results of statistical analysis obtained with principal component analysis (PCA) showed a major difference between the grassland honey and the other types of honey.

## 1. Introduction

According to Directive 2001/110 EC, honey is defined as the naturally occurring natural substance produced by *Apis melifera* bees [1]. Honey is a sweet substance naturally produced by the transformation and the processing of the nectar of bees or dew, which is stored in the cells of the honeycombs [2]. Honey has extremely varied composition due to geographical and botanical origin. Honey contains about 80% sugars [3,4] (fructose, glucose, maltose, sucrose, isomaltose, melezitose, raffinose, erlose, turanose, trehalose), and the other 20% is water, organic acids (acetic, butanoic, formic, citric, succinic, lactic, malic, pyroglutamic, gluconic, and a large number of aromatic acids) [5], amino acids (proline, asparagine acid and asparagine, calcium alginate, serine, methionine, tyrosine, leucine, lysine, arginine, histidine, ornithine, isoleucine, valine), minerals (calcium, iron, zinc, potassium, phosphorus, magnesium, selenium, chromium and manganese), group B vitamins (riboflavin, niacin, folic acid, pantothenic acid, pyridoxine and vitamin C), and enzymes (diastase, invertase, glucose oxidase, catalase) [6]. The sources of nectar are the spontaneous flora, the culture, and the sweet excretions of aphids or sweet and viscous substances that are secreted in certain periods by leaves and stems of trees. This diversity of sources results in a wide variety of types of honey with special chemical compositions, microbiological properties, and sensory qualities. Moreover, honey has anti-inflammatory properties, and its use as an antiviral, antibacterial, antiparasitic, antimutagenic, and anticancer agent has often been suggested [7,8,9,10,11]. Researchers proved that the properties of honey make it an important natural antioxidant due to its phenolic content [12,13]. In addition, correlations were reported between the color and the biological properties of honey, including phenolic content, antioxidant capacity, antimicrobial activity, and other parameters of honey [14,15]. Grassland honey has a special chemical composition due to the floral richness of the areas that act as sources of nectar and the soil composition. The grasslands are generally protected areas with dozens of flower species, where they come from the high content of polyphenols and their variety, transmitting to honey their special flavors and colors. The aim of this study was to draw attention to this type of honey and to promote efforts to keep this protected area from expansion of agricultural crops, which are treated with fertilizers and pesticides.

## 2. Results

Physicochemical parameters of six types of honey (rape, honeydew, linden, polyfloral, acacia, and grassland) were investigated. Table 1, Table 2 and Table 3 present the quality parameters (water content, color, electrical conductivity, viscosity, sugars, and polyphenols) of 14 rape, nine honeydews, 16 polyfloral, 21 lindens, 21 acacia, and 12 grassland honey samples. 

### 2.1. Water Content Determination

Water content determination is important because it influences the shelf-life of honey. High water content could accelerate crystallization in certain types of honey and increase its water activity to ferment and deteriorate its quality [16,17].

The Codex Alimentarius [18] standard specified that water content should not exceed 20% in honey to ensure safety against fermentation caused by the action of osmotolerant yeasts during storage [19].

High water content indicates extraction of a product in high humidity conditions or premature extraction. If the content is smaller, the honey crystallizes faster. Glucose/water content ratio (G/W) is another indicator for honey crystallization. If the G/W ratio is greater than 2.1, honey crystallizes faster [20]. All water content values in the tested honey samples were normal and did not exceed 20%. All results of water content were in accordance with the legislation established by the Codex [18].

The lowest water content was found in the honeydew (H) sample—H8 with 15.7%—and the highest content was found in acacia (A) honey and rape (R) honey, A11 and R9, respectively, with 18.0% each.

For rape honey and acacia honey, water content varied from 16.7% in R3 to 18.0% in R9, and from 16.3% in A20 to 18.0% in A11.

The values of water content for linden (L) honey samples ranged from 16.0% in L15 to 17.9% in L2 and L18. The water content of polyfloral (P) honey ranged from 15.9 % in P6 and P14 to 17.9% in P2. In honeydew and grassland (G) honeys, the water content varied from 15.7 % in H8 to 17.9% in H4 and from 16.3% in G3 to 17.6% in G1 and G6.

### 2.2. Electrical Conductivity (EC)

The electrical conductivity (EC) indicates the ability of honey to conduct an electric current, and tests were carried by ions and chemical modification in the honey. The Codex Alimentarius requires honey to have an electrical conductivity no greater than 0.8 mS/cm. The EC is an important physicochemical parameter in unifloral honey authentication.

This parameter is influenced by storage condition, temperature, water content, minerals, and ions content. The lowest level of electrical conductivity was found in rape honey (R1 with 0.11 mS/cm), and the highest level was 0.73 mS/cm in honeydew H4. The highest value of EC in rape samples analyzed was 0.19 mS/cm in R9, and the lowest was 0.11 mS/cm in R1.

For linden honey, EC varied between 0.55 mS/cm in L15 and 0.72 mS/cm in L2. EC content in acacia and polyfloral honey varied from 0.22 in A4 and A10 to 0.35mS/cm in A11 and from 0.28 mS/cm in P11 to 0.45 mS/cm in P12. In honeydew and grassland honey, EC ranged between 0.57 mS/cm in H8 to 0.73 mS/cm in H4 and between 0.24 mS/cm in G6 to 0.35 mS/cm in G12. Similar results were found by Alves et al. [21]. The conductivity value of linden honey (0.64 mS/cm) fell within the established limits of Directive 2001/110 EU (<0.8 mS/cm) [22].

### 2.3. Color

The color of honey is one of the most important indicators for consumers. Lightly colored honey is preferred oven dark honey. The color of honey is dependent on factors such as floral origin and nectar source. In the current study, differences were determined for grassland characteristics and the other assortments of honey.

The color parameters analyzed were L*, a*, b*, (L*—luminosity, a*—from red(+) to green(-), b*—from yelow(+) to blue(-)), the chrome (saturation)and the hue angle (H) were also calculated.

The L* parameter represents the lightness of honey, and the parameter ranged between 19.67 in honeydew and 52.61 in acacia honey.

The a* parameter represents the green compound (negative a* values), which was present in all samples of rape honey, acacia honey, and linden honey. Higher values of the green compound were found for rape honey (−2.19). The red parameter (positive a* values) was found in grassland, honeydew, and polyfloral honey. Higher values of the red parameter were found for honeydew (9.29).

The yellow parameter (positive b* values) was found in all samples and in all types of honey analyzed. These values ranged between 4.56 and 18.61 in rape honey.

### 2.4. Viscosity

Honey viscosity is another important parameter for the evaluation of state, fluidity, and crystallization.

The highest value of viscosity (14.73 Pa·s) was found in rape honey, and the lowest level (4.18 Pa·s) of this parameter was found in some samples of polyfloral honey.

In order to emphasize the difference between the six types of honey, we performed principal component analysis (PCA), and the results are presented in the biplot from Figure 1. The PCA method limited all data into two main components covering 83.82% of the variability (PC1-50.05% and PC2-33.76%). Analyzed parameters such as electrical conductivity, chroma, a*, and yellow index were grouped around the polyfloral honey and the honeydew honey. Water content and viscosity had higher values for rape honey, and hue angle, luminosity, and b* were more representative for acacia honey. Grassland and linden honey had almost medium values and are located in the center of the biplot.

### 2.5. Sugars

Nine sugars were investigated from 98 samples of six types of honey. The results obtained are shown in Table 2.

Glucose and fructose were the predominant compounds in the honey samples analyzed but were found in different ratios from one assortment to the next. Although the concentration of both sugars varies depending on the origin of the honey, it is generally expected that fructose will be found in a higher proportion than glucose [23,24].

Glucose to fructose ratio (G/F) should be smaller than 1.2 (this ratio is used to evaluate honey granulation, because glucose is less soluble in water than fructose) [25].

Honeys with a high G/F ratio would remain liquid for longer periods because of the modification of the saturated level of glucose by the presence of the larger amount of fructose.

In rape, honeydew, and some samples of grassland honey, the predominant sugar found was glucose, and in acacia, polyfloral, linden, and some samples of grassland honey, the higher concentration of sugar was represented by fructose.

The lowest level of glucose concentration was found in acacia sample A20 with 23.63 g/100 g honey, and the highest concentration was 42.55 g/100 g honey in rape sample R1. Fructose content ranged from 28.58 g/100 g honey in rape sample R1 to 45.98 g/100 g honey in acacia sample A21.

Others sugar identified were sucrose, trehalose, maltose, melezitose, raffinose, erlose, and turanose, which were found in lower concentrations than glucose and fructose. The sucrose content is an important parameter of the authentication of honey. The presence of a high level of sucrose in honey indicates adulteration with different syrups by harvesting the product before maturation, and this content can be reduced by the action of the invertase enzyme.

The highest levels of sucrose and trehalose were in P9 with 1.94 g/100 g honey and in A21 with 3.74 g/100 g honey, respectively, and in R2, sucrose was not detectable. The lowest concentration of trehalose was found in A1 (0.83 g/100 g honey).

Maltose, melezitose, erlose, turanose, and raffinose contents were not detectable in some samples of rape honey, linden honey, polyfloral honey, acacia honey, and grassland honey. The highest content of maltose was in linden sample L6 with 2.34 g/100 g honey. Erlose, turanose, and raffinose were found in higher quantities in A17 with 4.28 g/100 g honey, in P2 with 8.84 g/100 g honey, and in R9 with 0.43 g/100 g honey, respectively. The high content of melezitose (1.98 ± 0.42) in the linden honey was explained by the presence of honeydew honey. The color of linden honey is light yellow to amber, and in the case of the presence of honeydew honey (it is possible in linden forests), it becomes a greenish-gray color, demonstrated by the values of the parameters a* −0.49 ± 0.31 and b* 13.85 ± 0.69, which indicated the color sample in the field of yellow-yellow-green (a* negative, b* positive, quadrant II). Usually, linden honey is rich in maltose and contains about 1.2–1.9% of it [26,27]. A content of 0.20 ± 0.53 maltose, as was obtained in the analyzed samples, shows that the linden honey was not matured—a process in which the maltose is formed and the product darkens. At a ratio of 0.875, as indicated by the literature, honey crystallizes after three months; at a ratio of 0.76 ± 0.02 for the analyzed honey, it would crystallize over a longer period.

The G/F ratio was smaller than 1.0 for the analyzed acacia, polyfloral, and linden honeys. These honeys have a smaller tendency to crystallize. The higher ratio was for acacia and was equal to 0.51.

Some samples of the grassland honey had a G/F < 1.0, and some samples had a G/F ratio > 1. Rape samples and honeydew samples were characterized by a greater G/F ratio (about 1.48 for rape honey), indicating rapid crystallization.

A higher G/W ratio was found for rape honey (2.30), and a lower ratio was found for acacia honey (1.53).

The method for the reduction of variables, PCA, was performed also for the sugars content. The PCA method narrowed the data into two main components covering 74.90% of the variability, thus PC1 represented 54.04% of data, and PC2 represented 20.86%. The results are illustrated in Figure 2. It can be observed that acacia honey was predominately fructose and trehalose, and a slightly lower concentration was found in linden honey. In the opposite quadrant, the grassland and the honeydew honeys were characterized by melesitose, glucose content, G/W ratio, G/F ratio, and sucrose. Rape honey and linden honey were characterized by maltose and raffinose. Polyfloral honey was distinguished from other types of honey by its erlose and turanose contents. Regarding the grassland honey, there was no specific sugar for this; all the sugars were present in medium concentrations compared to the other honeys.

### 2.6. Phenolic Compounds

Polyphenols generally act as primary antioxidants, providing protection by removing free radicals, thus ending the reactions in the oxidative chain [28,29]. Honey phenolic profiles vary depending on geographical origin and floral source [30,31]. 

The phenolic compounds found in the honey analyzed included gallic acid, protocatechuic acid, 4-hydrxybenzoic acid, vanillic acid, chlorogenic acid, caffeic acid, p-coumaric acid, rosmarinic acid, myricetin, quercitin, luteolin, and kaempferol. The total polyphenols quantities were significant for grassland honey and honeydew and less significant for acacia and rape honey. The results of the twelve phenolic compounds analyzed from honeys are presented in Table 3. Figure 3 presents the type and the quantities of polyphenols from the honey samples and allows for an evaluation of the major compounds that contribute to color and flavor. 

The gallic acid was representative for polyfloral honey. Honeydew was rich in caffeic acid, protocatechuic acid, quercitin, and rosmarinic acid. Grassland honey was characterized by vanillic acid, 4-hydroxybenzoic acid, p-coumaric acid, myricetin, luteolin, and kaempferol chlorogenic acid. In accordance with specialty literature that previously proved this, after honeydew, grassland honey had the highest antioxidant activity. The high level of gallic acid was 5.23 mg/100 g honey for polyfloral honey and grassland honey. The maximum concentration of caffeic acid was found in honeydew (5.53 mg/100 g honey), as well as protocatechuic acid (13.32 mg/100 g honey), quercitin (21.21 mg/100 g honey), and rosmarinic acid (60.60 mg/100 g honey). The maximum concentration of vanillic acid was found in grassland honey (44.80 mg/100 g honey), in addition to 4-hydroxybenzoic acid, (4.77 mg/100 g honey), p-coumaric acid, (24.23 mg/100 g honey), myricetin, (2.67 mg/100 g honey), luteolin (0.91 mg/100 g honey), kaempferol (4.52 mg/100 g honey), and chlorogenic acid (4.67 mg/100 g honey). In rape samples were gallic acid, protocatechuic acid, 4-hydroxybenzoic acid, caffeic acid, vanillic acid, p-coumaric acid, chlorogenic acid, and 4-hydroxybenzoic acid was the dominant phenolic compound for these samples (0.21 mg/100 g honey). For acacia honey samples, p-coumaric acid was dominant (0.29 mg/100 g honey). The other phenolics found in acacia honeys were gallic acid, caffeic acid, vanillic acid, p-coumaric acid, and chlorogenic acid. The majority of linden honeys were represented by rosemarinic acid (4.67 mg/100 g honey), and luteolin and vanillic acid were not detectable. Rosemarinic acid (6.57 mg/100 g honey) was representative for polyfloral honeys.

PCA analysis regarding the polyphenols as active variables and the honey type as active observations was performed, and the two main components covering 90.30% of the variability were determined (PC1-71.27% and PC2-19.03%). From Figure 4, it can be observed that there was a significant difference between the six samples. Grassland honey was rich in p-coumaric acid, 4-hydroxybenzoic acid, kaempferol, luteolin, vanilic acid, and chlorogenic acids content. Honeydew was characterized by protocatechuic acid, caffeic acid, quercitin, myricetic, and rosmarinic acid. The total polyphenols content was found to be higher in honeydew samples. In the opposite quadrant, acacia, rape, and linden honey had lower concentrations of all polyphenols, and gallic acid was concentrated in polyfloral honey.

### 2.7. Microbiological Analysis

The values obtained for parameters SPC, total coliforms (TC), *Bacillus cereus*, yeasts, and molds for each sample of honey investigated as shown in Appendix A were within acceptable limits according to standards in force. Pathogenic microflora was not detectable for any type of honey analyzed. The quality of samples was superior in correlation with the values obtained for parameters SPC, TC, *Bacillus cereus*, yeasts, and molds. SPC values varied from <10 to 40 cfu/g. TC and *Bacillus cereus* were not detectable for any sample analyzed. The higher values of yeasts and molds were 20 cfu/g and 10 cfu/g, respectively. *Clostridium botulinum* was absent, which is very important because its presence in honey is dangerous for babies under one year old [32].

In conclusion, all samples had superior microbiological characteristics, but the grassland honey had the highest microbiological quality. The microbiological characteristics were correlated with the phenolic compound content. A higher amount of polyphenols meant a better microbiological stability.

Pearson correlation was performed in order to correlate the analyzed parameters (polyphenols, sugars, color parameters, conductivity, and water content) for all the samples. In the case of rape honey, a strong negative correlation was found between glucose content and total polyphenols (−0.958), and there was a strong positive correlation between glucose (0.956), trehalose (0.900), melezitose (0.950), and fructose content. Also, a perfect correlation was obtained between the erlose concentration and the vanilic acid concentration (1.000). Person correlation for honeydew showed a stronger positive relationship between total polyphenols and melezitose (0.912), fructose (0.832), trehalose (0.793), and maltose (0.720). Other positive correlations were obtained for chlorogenic acid and 4-hydroxybenzoic acid (0.992), erlose and 4-hydroxybenzoic acid (0.839), and conductivity and water content (0.921). Negative correlations resulted between total polyphenols and G/F ratio (−0.808) and glucose content (−0.828). Polyfloral honey indicated a positive correlation between total polyphenols and fructose (0.955), trehalose (0.952), and melezitose (0.919), as well as one between erlose and caffeic acid (0.746). There was a negative correlation in the case of gallic acid with chlorogenic acid (−0.723) and rosmarinic acid (−0.760) as well as total polyphenols and glucose (−0.955). Linden honey only evidenced strong correlations between sugar and polyphenols content (negative for glucose with fructose, trehalose, and melezitose, and positive for total polyphenols with these sugars). In contrast to the other types of honey, acacia had no correlation between total polyphenols content and sugar content. The results of the Pearson correlation for grassland honey indicated that, besides the correlations obtained for the other types of honey, there were correlations given by phenolic acids such as chlorogenic and 4-hydroxybenzoic acid (0.952), p-coumaric and 4-hydroxybenzoic acid (0.695), and protocatechuic and gallic acids (0.577).

## 3. Discussion

The study draws attention to the special qualities of grassland honey with its high content of polyphenols and microbiological characteristics. The important role of polyphenols in honey composition has been emphasized by all researchers in the field. For the first time, a well-defined link was demonstrated between the global honey structure and its biological activities [33]. Complex structures of proteins and polyphenols are involved in honey antioxidant capacity, antibacterial activity, and hydrogen peroxide production. According to M. Bucekova and co-authors, the polyphenols promote antimicrobial activity by producing H_2_O_2_ in their autoxidation and by influencing the Fenton reaction to create the reactive hydroxyl radicals [34,35]. The highest content of polyphenols and flavonoids with an average of 30.49 mg/100 g was determined to be in honeydew honey followed by grassland honey with an average of 21.50 mg/100 g. Similar to grassland honey, values of 20–40 mg/100 g were presented by Kuś and co-authors for heather honey, and values of 27.4 mg GAE/100 g were found by Bucekova and co-authors for wildflowers [14,35]. The values of total phenolic compounds (TPC) obtained for Romanian monofloral honey were smaller than those obtained by other authors. The difference is greater in the case of honeydew honey, as the analyzed samples had values within the range of 14.95–80.47 mg GAE/100 g, and in the case of honeydew honey analyzed by Marchitas and co-authors [36], the samples had values within the range of 35–130 mg GAE/100 g. The values of the total phenolic content were slightly lower than those obtained by Małgorzata D˙zugan et al. [37] of 34.59–75.49 mg GAE/100 g for the samples of honeydew honey from southeastern Poland (Podkarpacie, Poland) with the same climate. The variation in TPC was most likely due to differences in geographical location, harvesting time, and storage conditions. Honeydew honeys showed the highest value of total phenolic compounds (phenolic acids and flavonoids) of all the honey studied, followed by grassland honey, polyfloral honey, and linden honey, respectively, but these values were also lower than those reported by Ciucure et al. [38]. The TPCs of Irish multi-floral urban and rural honeys were within the range of 10–50 mg GAE/100 g of honey [39]. The results obtained for TPC in rural honey were similar to those obtained for grassland honey in northern Romania, within the range of 13.01–45.88 mg GAE/100 g of honey, with values higher than unifloral honey, most likely due to differences in geographical location. The grassland honey had the highest microbiological quality and was free of microorganisms in more than 33% of the samples, and the other samples were well below the allowed limit. Pathogenic microflora was not detectable for grassland honey as they were for other types of honey analyzed. All samples were analyzed in terms of the presence of *B. cereus*, which is widespread in nature and frequently isolated from soil and growing plants. The *Bacillus cereus* group can cause two different types of foodborne illness: the diarrhoeal type caused by enterotoxin(s) produced during vegetative growth of *B. cereus* in the small intestine, and the emetic type [40]. At the species level, the vast majority (>60%) of the isolates belonging to the genus *Bacillus* fall into the *B. cereus* group or in class *Bacillus subtilis*. According to ISO 7932:2004, the confirmatory stage does not enable the distinction of *B. cereus* from other closely related *Bacillus* species, such as *Bacillus anthracis*, *Bacillus thuringiensis*, *Bacillus weihenstephanensis*, and *Bacillus mycoides*. It is necessary to perform a motility test to differentiate *B. cereus* from the other *Bacillus species* when their presence is suspected. In our study, microorganisms were isolated from various types of honeys. Only the application of the spectrometric technique of Paweł Pomastowski and co-authors allowed for an unambiguous distinction between species/species groups, e.g., *B. subtilis* or *B. cereus* groups [41]. The method is very important because it allows for the isolation of *B. subtilis* spores, which enjoy GRAS (Generally Regarded As Safe) status from the U.S. Food and Drug Administration (FDA) and are included in the European Food Safety Authority (EFSA) list of Qualified Presumption of Safety (QPS) with applications in biomedicine and biotechnology (as oral vaccines, disinfectants, probiotics, or display systems) [42,43]. The grassland honey is a special honey sourced from the nectar in wild flowers that exist as hundreds of unique species in the meadows of hilly areas in the north of the Romania. These meadows are the borders of rural areas, where traditional farming is the main activity. Intensive agricultural practices, monocultures, and the use of pesticides and fertilizers change the destination of land and determine the decrease of these areas and the degradation of bees’ habitats. Promoting this type of honey due to its qualities also means issuing a warning about the essential role of bees as pollinators and in maintaining biodiversity and ecosystems, protecting crops, as well as their essential role in food production and food safety. The authors of the paper intend to continue studying this type of honey as well as other hive products such as wax, pollen, royal jelly, and propolis.

## 4. Materials and Methods

### 4.1. Materials

The honey assortments analyzed were acacia honey (A, *n* = 21), linden honey (L, *n* = 21), rape honey (R, *n* = 14), polyfloral honey (P, *n* = 16), honeydew honey (H, *n* = 9), and grassland honey (G, *n* = 12) and were purchased from authorized local producers from North Romania (Apicola Suceava trading company, Suceava, Romania). The reagents used were analytically pure from Fluka (Charlotte, NC, United States), Lachner (Neratovice, Czech Republic), Merck KGaA (Darmstadtcity, Germany), Scharlau (Barcelona, Spania), Sigma-Aldrich (Steinheim, Germany), Santa Cruz Biotechnology Inc. (Dallas, TX, USA), LGC Labor GmbH, todylaborotories.com. Hydrochloric acid 35% was purchased from Lachner (Neratovice, Czech Republic). Glacial acetic acid was purchased from Merck KGaA, Darmstadt, Germany. Agar plates, potato-dextrose agar (PDA), and Sabouroud agar were purchased by Sigma-Aldrich Inc., and agar Deoxycholate-Citrate Lactose (ADCL) was provided by todylaborotories.com. Sugar standards composed of d (+) glucose, d (-) fructose, d (+) sucrose, and d (+) trahalose dihydrate were provided by Santa Cruz Biotechnology Inc (Dallas, TX, USA), and mannose, maltose monohydrate, melezitose, raffinose pentahydrate, erlose, and d (+) turanose were purchased from LGC Labor GmbH (Augsburg, Germany).

Phenolic standards comprising gallic acid, protocatechuic acid, 4-hydrxybenzoic acid, vanilic acid, chlorogenic acid, caffeic acid, p-coumaric acid, rosmarinic acid, myricetin, quercitin, luteolin, and kaempferol were purchased by Sigma-Aldrich Inc., Germany. Methanol was provided by Fluka (Charlotte, NC, United States), and acetonitrile was provided by Scharlau (Barcelona, Spania).

### 4.2. Methods

#### 4.2.1. Water Content

The refractometer analysis (Refractometer RE40, Mettler Toledo, OH, United State of America) was based on the direct correlation between the refractive index and the solids content of honey, which allowed for the calculation of water content [1] according to the Harmonized Methods of the International Honey Commission [44].

#### 4.2.2. Sugars

Glucose, fructose, sucrose, trehalose, mannose, maltose, melezitose, raffinose, erlose, and turanose were analyzed by HPLC 10 AD VP Shimadzu (Shimadzu Corp., Kyoto, Japan) with an RI detector (Refractive Index Detector) in accordance with Bogdanov [44]. A column of 150 × 4.6 mm i.d. and particle size 3 μm was used to separate the compounds. Each analyzed sample consisted of 5 g of honey dissolved in 40 mL water. The samples were transferred in a volumetric flask of 100 mL and filled with water to the mark. The samples were filtered through a membrane of 0.45 μm and were collected in vials. The conditions for the analysis were: acetonitrile/water (75:25, *v*/*v*) as mobile phase, flow rate 1.0 mL/min, temperature 40 °C, volume of sample 10 μL. Standard solutions of 1 mg/mL were used for the calibration curve of each analyzed sugar. The calibration curves obtained for all sugars were characterized by a linear regression factor higher than 0.9980. The peak area obtained for each analyzed sugar was compared with standard sugar, allowing their quantitative determination. Sugar contents were expressed as g/100 g honey. Each result was the mean of triplicate determinations.

#### 4.2.3. Phenolic Compounds

Phenolic compounds were analyzed by HPLC 10 AD VP Shimadzu-Japan with a diode array detector. A column Alltech 250 × 4.6 mm i.d. and particle size of 5 μm was used for phenolic compounds separation. Acidified water with hydrochloric acid at pH equal to 2.00 was used. With this, a solution of 40% methanol was prepared. This was the solute for the honey. Each sample consisted of 1 g of honey dissolved in 5 mL of solute, which was shacked on the magnetic stirrer and filtered through a membrane of 0.45 μm [45]. The conditions for the analysis were: a flow rate of 1.0 mL/min, acetic acid 0.1% and acetonitrile as the two mobile phases, temperature of 30 °C, volume of sample 10 μL. Using standard solutions of 1 mg/mL for each phenolic compound, a calibration curve was obtained. The linear regression R^2^ was higher than 0.9976. The peak area obtained for each analyzed phenolic compound was compared with the standard, allowing for their quantitative determination of polyphenols at 280 and 320 nm wavelengths. Polyphenols contents were expressed as mg/100 g honey. Each result was the mean of triplicate determinations.

Phenolic compounds were analyzed by the Folin–Ciocalteu method according to Pontis. The compounds were dissolved with 5 g of honey in 20 mL of distilled water and transferred to 50 mL flasks and made up to the stock to form the stock solution. Then, 0.5 mL of the stock solution was mixed with 0.3 mL of Folin–Ciocalteu reagent and 2 mL of 15% sodium carbonate. Then, 5 mL was added to volumetric flasks and filled up to the mark with distilled water. The mixture was kept in the dark for 2 h, after which the absorbance was measured with the Perkin Elmer LAMBDA EZ201 UV-VIS Spectrophotometer (Waltham, MA, United States) at a wavelength of 798 nm [46]. Results were expressed as mg of gallic acid equivalents (GAE)/100 g of honey.

#### 4.2.4. Color was Determined by the Chromometer CR 410—Konica Minolta

CIE L*a*b* is a color space defined by the International Commission on Illumination (CIE) and it expresses color as tree values: L*—the lightness parameter; a*—the red/green parameter, with +a* indicating red and −a* indicating green; and b*—the yellow/blue parameter, with +b* indicating yellow and −b* indicating blue. The L*, a*, and b* coordinate axis defined the three dimensional CIE color space [47]. Chromameter was calibrated with a reference white porcelain tile (L* = 91.59; a* = 0.31; b* = 0.33) before the determinations. The illuminant used was D 65 [47]. The chroma parameter represented color saturation and varied from brilliant to bright. Hue angle was the color attribute by which the color was perceived [48].

#### 4.2.5. Electrical Conductivity

Electrical conductivity was determined with conductometer Mettler Toledo MPC 227 with thermostatic control. The conductivity was calculated using the formula [44]:S = K × G(1)
where K is the cell constant in cm^−1^ and G is the electrical conductance in mS.

#### 4.2.6. Viscosity

Viscosity was determined with the Brookfield viscometer. The results were expressed in Pa·s.

#### 4.2.7. Microbiologic Analysis

Microbiological analysis was performed according to ISO 4831:2006. Microbiology of food and animal feeding stuffs: horizontal method for the detection and enumeration of coliforms, ISO 18593:2018. Microbiology of food and animal feeding stuffs: horizontal method for sampling techniques from surfaces using contact plates and swabs, ISO 7932:2004. Microbiology of food and animal feeding stuffs: horizontal method for the enumeration of presumptive *Bacillus cereus*, ISO 21527:2008. Microbiology of food and animal feeding stuffs: horizontal method for the enumeration of yeasts and molds.

The samples analyzed were prepared by mixing 10 g of each honey sample with 90 mL of peptone saline solution (8.5 g/L) to achieve the initial dilution. The mixture was then used for the other dilutions. All colonies appearing after incubation were counted. Microbial counts were expressed in colony forming units/gram of honey (cfu/g).
Number of standard counts. Serial dilutions from 10^−1^ to 10^−3^ of the initial dilution inoculated on agar plates were recommended. The plates were incubated for 48 h at 30 °C.Bacillus. The initial dilution was brought to 80 °C and held at this temperature for 10 min, after which it was cooled rapidly in ice. Bacteria for aerobic spore formation were inoculated on agar medium. The plates were incubated for 48 h at 30 °C.Number of coliforms. Inoculation was carried out on Agar Deoxycholate-Citrate Lactose medium (ADCL- todylabrotaories.com). The plates were incubated for 24 h at 37 °C.Yeast. From the initial dilution, they were inoculated on agar potato-dextrose medium (PDA) and incubated for 72 h at 25 °C.Molds. The mold was inoculated on Sabouroud agar medium and incubated for 7 days at 25 °C [20].

### 4.3. Statistical Analysis

The resultant data were analyzed by PCA, Pearson correlation, and descriptive analyses with XLSTAT software (2018, trial version, Addinsoft, NY, United States). PCA evaluated the correlations between the honey type, and the physico-chemical parameters analyzed the variation and extracted the main components.

## 5. Conclusions

Physicochemical and microbiological parameters were found to be in the required standards for all honey samples. High content of polyphenols was found in grassland honey and in honeydew. Grassland honey is characterized not only by the high content of polyphenols but also by the presence of all phenolic compounds analyzed.

## Figures and Tables

**Figure 1 molecules-24-02932-f001:**
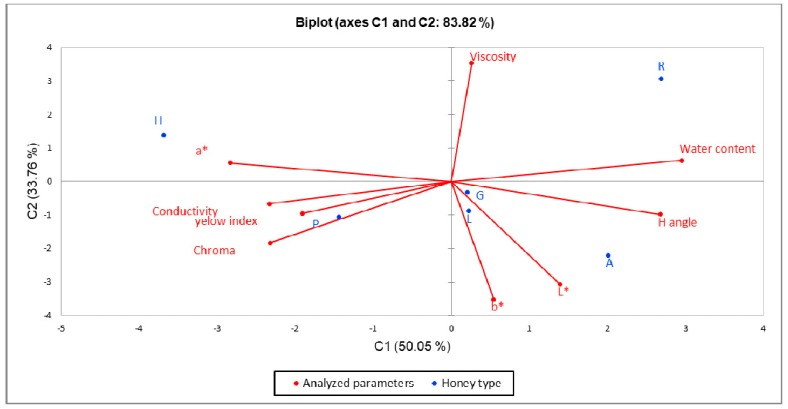
Principal component analysis (PCA) of the dataset consisting of analyzed parameters of each honey sample (R-rape, H-honeydew, P-polyfloral, L-linden, A-acacia, G-grassland).

**Figure 2 molecules-24-02932-f002:**
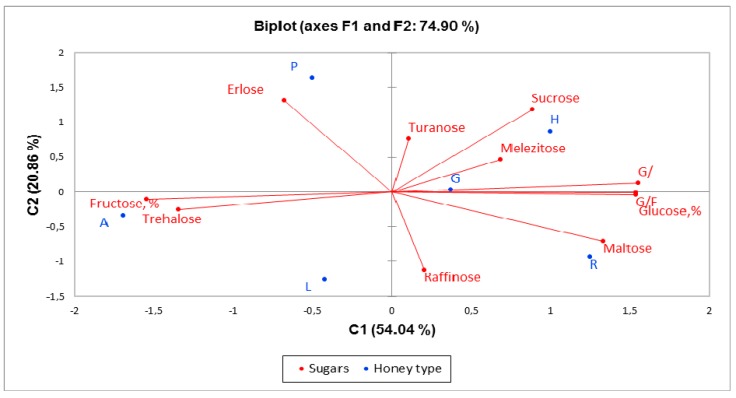
Principal component analysis of the dataset consisting of sugar concentrations of each honey sample. (R-rape, H-honeydew, P-polyfloral, L-linden, A-acacia, G-grassland).

**Figure 3 molecules-24-02932-f003:**
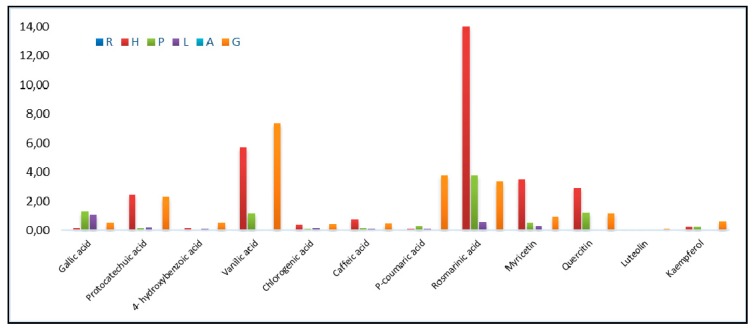
Quantities of polyphenols in honey samples (mg/100 g).

**Figure 4 molecules-24-02932-f004:**
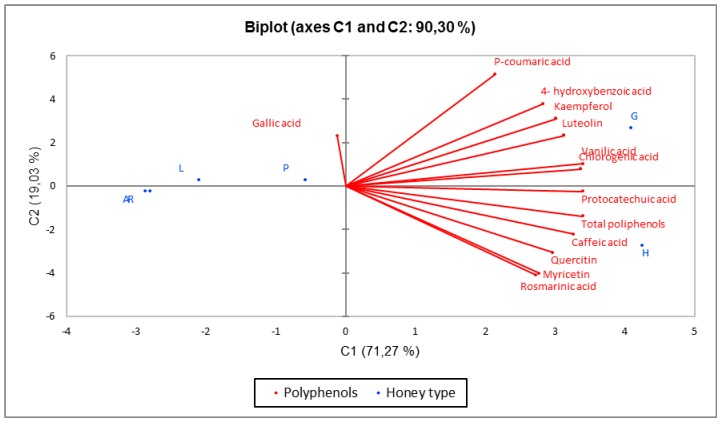
Principal component analysis of the dataset consisting of polyphenols concentration of each honey sample (R-rape, H-honeydew, P-polyfloral, L-linden, A-acacia, G-grassland).

**Table 1 molecules-24-02932-t001:** Physicochemical parameters of different types of honey from Romania.

Honey Variety		Water Content, %	L*	a*	b*	Chroma	Hue angle, (degrees)	Yellow Index	Conductivity (mS/cm)	Viscosity (Pa·s)
Rape (n = 14)	Mean ± SD	17.32 ± 0.45	26.06 ± 5.55	−0.72 ± 0.64	7.83 ± 4.12	3.51 ± 0.93	84.99 ± 3.36	42.48 ± 6.05	0.14 ± 0.02	12.69 ± 1.12
	Min	16.70	17.61	−2.19	4.56	2.66	76.77	25.03	0.11	10.96
	Max	18.00	38.07	−0.14	18.61	5.79	88.81	78.93	0.19	14.73
Honeydew (n = 9)	Mean ± SD	16.62 ± 0.71	21.71 ± 1.38	7.06 ± 1.20	8.36 ± 1.45	5.53 ± 0.38	49.72 ± 5.55	55.24 ± 10.10	0.64 ± 0.05	8.46 ± 0.74
	Min	15.70	19.67	5.35	6.17	4.80	41.82	36.97	0.57	7.14
	Max	17.90	23.84	9.29	10.24	5.94	57.29	69.40	0.73	9.47
Polyfloral (n = 16)	Mean ± SD	16.78 ± 0.65	36.13 ± 2.16	4.69 ± 0.74	13.77 ± 0.57	5.87 ± 0.22	71.26 ± 2.51	54.67 ± 4.43	0.38 ± 0.05	6.63 ± 1.30
	Min	15.90	31.90	3.24	12.74	5.46	66.28	48.87	0.28	4.18
	Max	17.90	39.65	5.98	14.82	6.21	76.39	62.14	0.45	8.62
Linden (n = 21)	Mean ± SD	17.01 ± 0.51	33.06 ± 1.76	−0.49 ± 0.31	13.85 ± 0.69	4.39 ± 0.21	87.96 ± 1.32	60.02 ± 4.50	0.61 ± 0.04	6.37 ± 0.96
	Min	16.00	27.92	−1.21	12.43	4.02	85.05	51.29	0.55	4.76
	Max	17.90	35.27	−0.12	14.94	4.84	89.52	70.20	0.72	8.06
Acacia (n = 21)	Mean ± SD	17.13 ± 0.48	49.26 ± 2.23	−1.11 ± 0.32	15.05 ± 1.07	4.92 ± 0.17	85.73 ± 1.30	43.77 ± 4.01	0.28 ± 0.04	3.35 ± 0.65
	Min	16.30	45.28	−1.69	13.25	4.56	83.20	35.98	0.22	2.12
	Max	18.00	52.61	−0.45	16.38	5.29	88.43	51.27	0.35	4.27
Grassland (n = 12)	Mean ± SD	17.03 ± 0.38	32.77 ± 1.94	1.81 ± 1.74	13.31 ± 0.78	4.87 ± 0.52	82.22 ± 7.46	58.15 ± 3.68	0.31 ± 0.04	7.71 ± 0.72
	Min	16.30	29.34	0.31	12.15	4.35	62.63	52.04	0.24	6.39
	Max	17.60	35.23	6.29	14.59	5.99	88.78	63.39	0.35	8.64

SD: standard deviation; Min: minimum value; Max: maximum value. L*: lightness of honey, a*—from red(+) to green (-), b*—from yelow(+) to blue(-), chroma-saturation.

**Table 2 molecules-24-02932-t002:** Sugar content of different types of honey from North Romania.

Honey Variety		Sugars (g/100 g)										
		Glucose	Fructose	Sucrose	Trehalose	Melezitose	Maltose	Erlose	Turanose	Raffinose	G/F	G/W
Rape (n = 14)	Mean ± SD	39.95 ± 1.54	30.26 ± 1.06	0.93 ± 0.66	0.01 ± 0.00	0.01 ± 0.00	0.29 ± 0.43	0.00 ± 0.02	0.17 ± 0.13	0.05 ± 0.12	1.31 ± 0.09	2.30 ± 0.12
	Min.	38.02	28.58	0.00	0.00	0.00	0.00	0.00	0.01	0.00	1.20	2.16
	Max.	42.55	31.54	1.76	0.02	0.02	1.24	0.10	0.44	0.43	1.48	2.49
Honeydew (n = 9)	Mean ± SD	36.41 ± 0.54	32.41 ± 0.19	1.36 ± 0.28	0.04 ± 0.00	4.89 ± 1.56	0.27 ± 0.33	0.42 ± 0.53	0.10 ± 0.06	0.01 ± 0.01	1.12 ± 0.02	2.19 ± 0.09
	Min.	35.68	32.14	0.74	0.03	1.24	0.00	0.00	0.03	0.00	1.09	2.03
	Max.	37.21	32.68	1.80	0.04	5.84	0.76	1.54	0.20	0.02	1.15	2.34
Polyfloral (n = 16)	Mean ± SD	31.96 ± 0.26	38.55 ± 0.61	1.12 ± 0.57	0.25 ± 0.02	0.71 ± 0.14	0.01 ± 0.02	1.23 ± 2.66	0.23 ± 0.20	0.03 ± 0.01	0.82 ± 0.02	1.90 ± 0.07
	Min.	31.62	37.54	0.19	0.23	0.58	0.00	0.00	0.00	0.01	0.80	1.79
	Max.	32.31	39.35	1.95	0.28	1.12	0.08	8.85	0.71	0.05	0.86	2.02
Linden (n = 21)	Mean ± SD	30.88 ± 0.41	40.48 ± 0.53	0.51 ± 0.44	0.32 ± 0.03	1.98 ± 0.42	0.20 ± 0.53	0.08 ± 0.16	0.11 ± 0.12	0.05 ± 0.09	0.76 ± 0.02	1.81 ± 0.06
	Min.	30.30	39.43	0.08	0.28	1.13	0.00	0.00	0.00	0.00	0.73	1.70
	Max.	31.56	41.15	1.67	0.38	2.56	2.34	0.68	0.40	0.38	0.80	1.92
Acacia (n = 21)	Mean ± SD	26.21 ± 1.62	44.11 ± 1.03	0.63 ± 0.51	1.85 ± 0.95	0.04 ± 0.01	0.05 ± 0.14	0.53 ± 1.15	0.12 ± 0.14	0.02 ± 0.03	0.59 ± 0.05	1.53 ± 0.10
	Min.	23.63	42.65	0.04	0.83	0.03	0.00	0.00	0.00	0.00	0.51	1.32
	Max.	28.26	45.98	1.58	3.74	0.07	0.62	4.28	0.45	0.13	0.66	1.66
Grassland (n = 12)	Mean ± SD	36.82 ± 1.91	36.64 ± 3.66	0.60 ± 0.28	0.29 ± 0.20	1.88 ± 1.36	0.18 ± 0.31	0.07 ± 0.15	0.19 ± 0.16	0.02 ± 0.01	1.02 ± 0.14	2.16 ± 0.12
	Min.	34.26	31.85	0.27	0.03	0.03	0.00	0.00	0.01	0.00	0.83	1.99
	Max.	39.98	41.23	0.99	0.47	3.11	0.87	0.52	0.44	0.04	1.19	2.34

SD: standard deviation; Min: minimum value; Max: maximum value. G/F: glucose to fructose ratio; G/W: glucose/water content ratio.

**Table 3 molecules-24-02932-t003:** The results of the concentration of the twelve phenolic compounds.

Honey Variety		Phenolic Compound (mg/100 g)													
		Gallic Acid	Protocatechuic Acid	4-Hydroxy Benzoic Acid	Vanilic Acid	Chlorogenic Acid	Caffeic Acid	P-Coumaric Acid	Rosmarinic Acid	Myricetin	Quercitin	Luteolin	Kaempferol	Total Poliphenols	Total Poliphenols Content, mg GAE/100 g
Rape (n = 14)	Mean ± SD	0.00 ± 0.01	0.01 ± 0.05	0.01 ± 0.03	0.01 ± 0.03	0.02 ± 0.05	0.04 ± 0.05	0.03 ± 0.05	0.00 ± 0.00	0.00 ± 0.00	0.00 ± 0.00	0.00 ± 0.00	0.00 ± 0.00	0.14 ± 0.04	0.17 ± 0.02
	Min	0.00	0.00	0.00	0.00	0.00	0.00	0.00	0.00	0.00	0.00	0.00	0.00	0.08	0.12
	Max	0.07	0.21	0.14	0.14	0.17	0.15	0.16	0.00	0.00	0.00	0.00	0.00	0.21	0.22
Honeydew (n = 9)	Mean ± SD	0.14 ± 0.22	2.44 ± 4.62	0.18 ± 0.37	5.74 ± 9.19	0.37 ± 0.65	0.74 ± 1.80	0.12 ± 0.09	14.01 ± 23.95	3.47 ± 6.47	2.91 ± 6.95	0.08 ± 0.24	0.27 ± 0.40	30.49 ± 22.13	32.5 ± 0.8
	Min	0.00	0.00	0.00	0.00	0.00	0.00	0.00	0.00	0.00	0.00	0.00	0.00	14.87	14.95
	Max	0.61	13.32	0.99	24.81	1.78	5.53	0.23	60.60	19.75	21.21	0.71	1.18	80.38	80.47
Polyfloral (n = 16)	Mean ± SD	1.27 ± 2.10	0.15 ± 0.24	0.08 ± 0.18	1.20 ± 1.03	0.13 ± 0.31	0.14 ± 0.45	0.29 ± 0.55	3.80 ± 2.37	0.50 ± 0.85	1.23 ± 1.89	0.07 ± 0.28	0.24 ± 0.35	9.13 ± 1.84	9.36 ± 1.82
	Min	0.00	0.00	0.00	0.00	0.00	0.00	0.00	0.00	0.00	0.00	0.00	0.00	6.54	6.78
	Max	5.23	0.65	0.52	3.10	1.16	1.84	2.32	6.57	2.26	6.13	1.12	1.21	12.81	12.94
Linden (n = 21)	Mean± SD	1.10 ± 1.28	0.22 ± 0.43	0.10 ± 0.19	0.00 ± 0.00	0.15 ± 0.47	0.10 ± 0.28	0.12 ± 0.29	0.57 ± 1.36	0.31 ± 0.59	0.03 ± 0.10	0.00 ± 0.00	0.01 ± 0.04	2.71 ± 1.64	2.83 ± 1.63
	Min	0.00	0.00	0.00	0.00	0.00	0.00	0.00	0.00	0.00	0.00	0.00	0.00	0.94	1.06
	Max	3.97	1.72	0.54	0.00	2.15	1.20	1.25	4.67	1.92	0.48	0.00	0.20	5.44	5.49
Acacia (n = 21)	Mean± SD	0.02 ± 0.05	0.00 ± 0.00	0.00 ± 0.01	0.00 ± 0.02	0.01 ± 0.03	0.03 ± 0.03	0.02 ± 0.06	0.00 ± 0.00	0.00 ± 0.00	0.00 ± 0.00	0.00 ± 0.00	0.00 ± 0.00	0.08 ± 0.06	0.09 ± 0.05
	Min	0.00	0.00	0.00	0.00	0.00	0.00	0.00	0.00	0.00	0.00	0.00	0.00	0.02	0.04
	Max	0.20	0.00	0.02	0.09	0.14	0.09	0.29	0.00	0.00	0.00	0.00	0.00	0.28	0.32
Grassland (n = 12)	Mean ± SD	0.53 ± 1.50	2.31 ± 3.52	0.52 ± 1.40	7.35 ± 12.48	0.41 ± 1.34	0.44 ± 1.08	3.78 ± 8.60	3.35 ± 2.81	0.92 ± 1.10	1.16 ± 1.52	0.14 ± 0.32	0.60 ± 1.27	21.50 ± 10.73	22.16 ± 11.01
	Min	0.00	0.00	0.00	0.00	0.00	0.00	0.00	0.00	0.00	0.00	0.00	0.00	12.97	13.01
	Max	5.23	8.85	4.78	44.81	4.67	3.56	24.23	6.23	2.67	4.26	0.92	4.53	45.85	45.88

SD: standard deviation; Min: minimum value; Max: maximum value.

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
