# Peer review of "Comparison of Physicochemical, Microbiological Properties and Bioactive Compounds Content of Grassland Honey and other Floral Origin Honeys"

_molecules, 2019, doi:10.3390/molecules24162932_

Round 1
Reviewer 1 Report
Review of the article: „Comparison of physicochemical, microbiological properties and bioactive compounds content of grassland honey and other floral origin honeys
Manuscript ID: molecules-550946
The botanical source of the nectar (or honeydew) is crucial for honey taste, colour but also biological activity of this product. The proposed manuscript is well prepared and interesting from both scientific but also practical points of view. All experiments were well planned and performed. Important advantage of the manuscript is the fact that the authors investigated large group of products – honeys produced from the nectar of different species of plants and honeydew. The obtained results are interesting and worth of publication. However, I have some remarks and comments which the authors should take in into account preparing the revised version of the publication. The detailed comments are presented below.
Abstract
In my opinion the abstract should be changed. In the current version the authors only present some general comments, e.g.: “The total polyphenols content is significant for grassland honey and honeydew, and less significant for acacia and rape honey. All samples are microbiologically safe but the grassland honey has the highest microbiological quality”. In my opinion some values (concrete results) of the most important determinations should be presented in the abstract.
Introduction
Lines 41- 42 – antimicrobial (especially antibacterial) activity of honey should be mentioned in this place
Lines 44-45 – I am not sure that high water content is important for high antioxidant activity – please check it. Moreover, some authors (e.g. Kuś et al., found correlation between colour of honey and biological properties of this product, including phenolic content, antioxidant capacity, antimicrobial potential and other parameters - doi: 10.1111/lam.12541.)
Results
Line 63 – instead of “Moisture” I would propose using “Water content”
Line 71 – please define G/M
Some of the results are interesting and surprising. First of all the high values of conductivity od content of melezitose suggest that honeys collected from linden trees contain also some amount of honeydew (aphid secretion). I am also surprized with low content of melezitose in honeydew honeys and high concentration of this sugar in acacia honey, interesting is also the ratio of glucose and fructose in linden honeys. I would be grateful for a short comment.
Lines 126-127 Figure 1 – the description/legend under the picture should be a bit more detailed (at least it should be clearly written what do R, A, G, L, H mean. The same comment about Figures 2 and 3.
Line 186 – I regret, that the authors did not determined total content of phenolic compounds using Folin-Ciocalteu method.
Line 228 – Microbial analysis – important advantage of the manuscript. Not many publication about microbial contamination of honey are available.
Discussion is very poor – the most important weakness of the manuscript (other remarks are minor of importance). The obtained results, especially content of phenolic compounds should be compared with results presented by other authors. Interesting analysis in this area has been recently presented by several research groups in Molecules journal, e.g. Grecka et al., https://doi.org/10.3390/molecules23020260, Bucekowa et al., https://doi.org/10.3390/molecules24081573, but also many other authors in other journals. Moreover, the biological role of these compounds should be discussed. For this purpose I would especially recommend the publications of the group of Prof Brudzynski – in my opinion this research group have presented the most advanced studies of the role of polyphenols for the biological activity of honey (doi: 10.3390/molecules23123067.; doi: 10.1038/s41598-017-08072-0.)
As I noticed above, to date not many publications about microbial contamination of honey are available. Thus, the results presented by the authors are very important. But, the authors should compare the results of their investigation with results presented by other authors. It also should be noticed that recently some authors revealed that isolates of bacteria recovered from honey (mostly of the genus Bacillus) often exhibit ability to production of antimicrobial agents – it also should be mentioned in discussion.
Materials and methods
All experiments were well planned and perform, I only regret that total phenolic content was not determined.
Final decision – minor revision (the authors should prepare more detailed discussion, other remarks are of minor importance).
Author Response
Thank you for your comments and suggestions, we modified it as you suggested.

Reviewer 2 Report
1. The manuscript contains scientific data about various varieties of honey from northern Romania.
2. The manuscript presents scientific data but they are not discussed in comparison with similar scientific data.
3. Table 4 may be removed or presented as additional data because the information is repeated in the text.
4. Section 3 should provide comparative scientific data.
5. In section 4.2.3, the samples were diluted 50 times. It is well known that honey contains small amounts of polyphenols. What were the limits of detection and quantification of the HPLC method, and how did you eliminate the interference given by sugars? Can you present a figure with polyphenols separation?
6. I do not think it's right what you said here: ‘’the peak area obtained for each analyzed sugar is 325 compared with standard sugar allowing their quantitative determination of polyphenols at 280 and 326 320 nm wavelength’’.
Author Response

(The authors gave the same response as above.)

Reviewer 3 Report
The manuscript measures multiple properties of six kinds of honey, in an effort to draw attention to Grassland honey. The research results in a good amount of data, but still need improvement to clearly present & demonstrate the specialty of Grassland honey.
Major:
1. Need simplified tables and bar plots to focus the presentation.
Table 1, 2 and 3 are overwhelming with data. Consider simplifying them by combining Mean and SD into X±Y format, and put the more complete tables to supplemental.
I would also suggest presenting bar plots, but it's not feasible for so many properties in this manuscript. Authors may still consider selectively present bar plots of several important properties in main text (for example, to show the high polyphenols content in grassland honey), and put the rest in supplemental.
Table 4 details every honey sample, which apparently isn't suitable for main text. This belongs to supplemental. Also, you may want to replace "Absent" with a simplified sign such as dash "-" and put a footnote. Use a simplified table or plot for main text if necessary.
2. Double check data.
Some data is apparently wrong: Table 2, Rape G/M (mean 2.30, min 2.16, max 2.16). Please double check all data!
Minor:
1. A table in "Supplementary files.docx" is overflowed.
2. Line 50: "maintain ... by ...", do you mean "keep ... from ..."?
3. Confusion of *, • and ×
Line 339, use multiply sign: K × G
Table 1 & Line 343, use dot for the unit: Pa•s
Line 346, use either g/L or g•L-1
4. Abbreviations:
Abbreviations should be explained upon first occurrence. For example: line 71, G/M.
Single letter abbreviations (for example, G for Grassland honey) for different kinds of honey are not explained in PCA Figures.
5. Figure 1, 2 and 3, the comma in percentage should be dot.
Author Response

(The authors gave the same response as above.)

Round 2
Reviewer 2 Report
I agree that the changes made have improved the quality of the manuscript. However, when I referred to the discussions on the results I would like to see references to other results (polyphenols contents, type of polyphenols, physiochemical properties,...) obtained for the different kinds of honey coming from the same country or from neighbours (with same climate). Here are some articles that provide information about them: Mircea Oroian, Sorina Ropciuc, Sergiu Paduret, Elena Todosi Sanduleac, Authentication of Romanian honeys based on physicochemical properties, texture and chemometric, J Food Sci Technol (December 2017) 54(13):4240–4250, DOI 10.1007/s13197-017-2893-0 Corina Teodora Ciucure, Elisabeta‐Irina Geană, Phenolic compounds profile and biochemical properties of honeys in relationship to the honey floral sources, Phytochemical Analysis. 2019;1–12, DOI: 10.1002/pca.2831 Marghitas Liviu Al , Dezmirean Daniel , Adela Moise, Otilia Bobis, Laura Laslo, Stefan Bogdanov, Physico-chemical and bioactive properties of different floral origin honeys from Romania, Food Chemistry 112 (2009) 863–867, doi:10.1016/j.foodchem.2008.06.055 Raluca Popescu, Elisabeta Irina Geana,Oana Romina Dinca,Claudia Sandru,Diana Costinel&Roxana Elena Ionete, Characterization of the Quality and Floral Origin of Romanian Honey, DOI: 10.1080/00032719.2015.1057830Author Response
Please see the attachment.
